# ONE-SHOT IMITATION FROM OBSERVING HUMANS VIA DOMAIN-ADAPTIVE META-LEARNING

**Tianhe Yu\*, Chelsea Finn\*, Annie Xie, Sudeep Dasari, Tianhao Zhang,**
**Pieter Abbeel, Sergey Levine**
University of California, Berkeley
\* denotes equal contribution

## 1 INTRODUCTION

Demonstrations provide a descriptive medium for specifying robotic tasks. Prior work has shown that robots can acquire a range of complex skills through demonstration, such as table tennis (Mülling et al., 2013), lane following (Pomerleau, 1989), pouring water (Pastor et al., 2009), drawer opening (Rana et al., 2017), and multi-stage manipulation tasks (Zhang et al., 2018). However, the most effective methods for robot imitation differ significantly from how humans and animals might imitate behaviors: while robots typically need to receive demonstrations in the form of kinesthetic teaching (Pastor et al., 2011; Akgun et al., 2012) or teleoperation (Calinon et al., 2009; Rahmatizadeh et al., 2017; Zhang et al., 2018), humans and animals can acquire the gist of a behavior simply by *watching* someone else. In fact, we can adapt to variations in morphology, context, and task details effortlessly, compensating for whatever *domain shift* may be present and recovering a skill that we can use in new situations (Brass & Heyes, 2005). Additionally, we can do this from a very small number of demonstrations, often only one. How can we endow robots with the same ability to learn behaviors from raw third person observations of human demonstrators?

Acquiring skills from raw camera observations presents two major challenges. First, the difference in appearance and morphology of the human demonstrator from the robot introduces a systematic domain shift, namely the correspondence problem (Nehaniv et al., 2002; Brass & Heyes, 2005). Second, learning from raw visual inputs typically requires a substantial amount of data, with modern deep learning vision systems using hundreds of thousands to millions of images (Xiang et al., 2017; Kim & Walter, 2017). In this paper, we demonstrate that we can begin to address both of these challenges through a single approach based on meta-learning. Instead of manually specifying the correspondence between human and robot, which can be particularly complex for skills where different morphologies require different strategies, we propose a data-driven approach. Our approach can acquire new skills from only one video of a human. To enable this, it builds a rich prior over tasks during a *meta-training* phase, where both human demonstrations and teleoperated demonstrations are available for a variety of other, structurally similar tasks. In essence, the robot learns how to learn from humans. After the meta-training phase, the robot can acquire new skills by combining its learned prior knowledge with one video of a human performing the new skill.

The main contribution of this paper is a system for learning robotic manipulation skills from a single video of a human by leveraging large amounts of prior meta-training data, collected for different tasks. When deployed, the robot can adapt to a particular task with novel objects using just a single video of a human performing the task with those objects. The video of the human need not be from the same perspective as the robot, or even be in the same room. The robot is trained using videos of humans performing tasks with various objects along with demonstrations of the robot performing the same task. Our experiments on two real robotic platforms demonstrate the ability to learn directly from RGB videos of humans, and to handle novel objects, novel humans, and videos of humans in novel scenes. The full version of this paper is available online.[1]

## 2 DOMAIN-ADAPTIVE META-LEARNING

We develop a domain-adaptive meta-learning method, which will allow us to handle the setting of learning from video demonstrations of humans. Our approach consists of two phases. First, in the meta-training phase, the goal will be to acquire a prior over policies using both human and robot demonstration data, that can then be used to quickly learn to imitate new tasks with only human demonstrations. For meta-training, we will assume a distribution over tasks $p(\mathcal{T})$, a set of tasks $\{\mathcal{T}_i\}$ drawn from $p(\mathcal{T})$ and, for each task, two small datasets containing several human and robot

---

[1] The full paper is at `https://arxiv.org/abs/1802.01557`

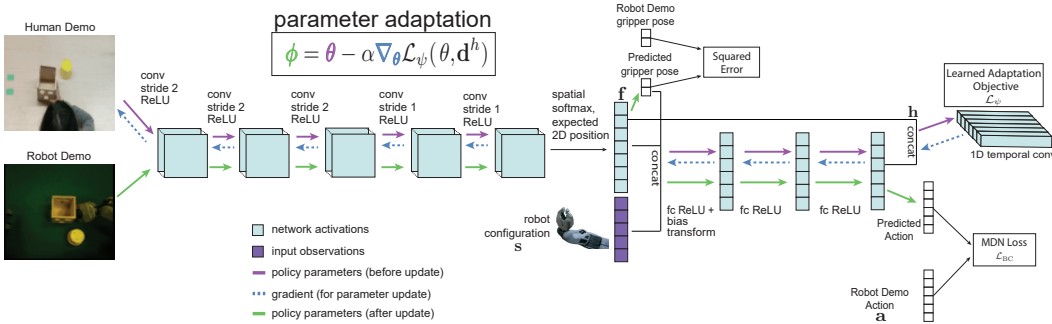

Figure 1: Illustration of the policy architecture. The policy consists of a sequence of five convolutional (conv) layers, followed by a spatial soft-argmax and fully-connected (fc) layers.

demonstrations, respectively: $(\mathcal{D}_{\mathcal{T}_i}^h, \mathcal{D}_{\mathcal{T}_i}^r)$. After the meta-training phase, the learned prior can be used in the second phase, when the method is provided with a human demonstration of a new task $\mathcal{T}$ drawn from $p(\mathcal{T})$. The robot must combine its prior with the new human demonstration to infer policy parameters $\phi_{\mathcal{T}}$ that solve the new task. While we will extend the model-agnostic meta-learning (MAML) (Finn et al., 2017a) algorithm for this purpose, the key idea of our approach is applicable to other meta-learning algorithms. Like the MAML algorithm, we will learn a set of initial parameters, such that after one or a few steps of gradient descent on just one human demonstration, the model can effectively perform the new task. Thus, the data $\mathcal{D}_{\mathcal{T}}^{\text{tr}}$ will contain one human demonstration of task $\mathcal{T}$, and the data $\mathcal{D}_{\mathcal{T}}^{\text{val}}$ will contain one or more robot demonstrations of the same task.

Unfortunately, we cannot use a standard imitation learning loss for the inner adaptation objective computed using $\mathcal{D}_{\mathcal{T}}^{\text{tr}}$, since we do not have access to the human's actions. Even if we knew the human's actions, they will typically not correspond directly to the robot's actions. Instead, we propose to meta-learn an adaptation objective that does not require actions, and instead operates only on the policy activations. During the meta-training phase, we will learn both an initialization $\theta$ and the parameters $\psi$ of the adaptation objective $\mathcal{L}_{\psi}$ which will operate

**Algorithm 1** Domain-Adaptive Meta-Learning

**Require:** $\{(\mathcal{D}_{\mathcal{T}_i}^h, \mathcal{D}_{\mathcal{T}_i}^r)\}$: human and robot demonstration data for a set of tasks $\{\mathcal{T}_i\}$
  **while** training **do**
    Sample task $\mathcal{T} \sim \{\mathcal{T}_i\}$ {or minibatch}
    Sample video of human $\mathbf{d}^h \sim \mathcal{D}_{\mathcal{T}}^h$
    Compute policy parameters:
      $\phi_{\mathcal{T}} = \theta - \alpha \nabla_{\theta} \mathcal{L}_{\psi}(\theta, \mathbf{d}^h)$
    Sample robot demo $\mathbf{d}^r \sim \mathcal{D}_{\mathcal{T}}^r$
    $(\theta, \psi) \leftarrow (\theta, \psi) - \beta \nabla_{\theta, \psi} \mathcal{L}_{\text{BC}}(\phi_{\mathcal{T}}, \mathbf{d}^r)$
  **end while**
  Return $\theta, \psi$

only on the activations of the policy. The parameters $\theta$ and $\psi$ are optimized for choosing actions that match the robot demonstrations in $\mathcal{D}_{\mathcal{T}}^{\text{val}}$. After meta-training, the parameters $\theta$ and $\psi$ are retained, while the data is discarded. A human demonstration $\mathbf{d}^h$ is provided for a new task $\mathcal{T}$ (but not a robot demonstration). To infer the policy parameters for the new task, we use gradient descent starting from $\theta$ using the learned loss $\mathcal{L}_{\psi}$ and one human demonstration $\mathbf{d}^h$: $\phi_{\mathcal{T}} = \theta - \alpha \nabla_{\theta} \mathcal{L}_{\psi}(\theta, \mathbf{d}^h)$.

We optimize for task performance during meta-training using a behavioral cloning objective that maximizes the probability of the expert actions in $\mathcal{D}^{\text{val}}$. In particular, for a policy parameterized by $\phi$ that outputs a distribution over actions $\pi_{\phi}(\cdot | \mathbf{o}, \mathbf{s})$, the behavioral cloning objective is $\mathcal{L}_{\text{BC}}(\phi, \mathbf{d}^r) = \mathcal{L}_{\text{BC}}(\phi, \{\mathbf{o}_{1:T}, \mathbf{s}_{1:T}, \mathbf{a}_{1:T}\}) = \sum_t \log \pi_{\phi}(\mathbf{a}_t | \mathbf{o}_t, \mathbf{s}_t)$ Putting this together with the inner gradient descent adaptation, the meta-training objective is the following:

$$\min_{\theta, \psi} \sum_{\mathcal{T} \sim p(\mathcal{T})} \sum_{\mathbf{d}^h \in \mathcal{D}_{\mathcal{T}}^h} \sum_{\mathbf{d}^r \in \mathcal{D}_{\mathcal{T}}^r} \mathcal{L}_{\text{BC}}(\theta - \alpha \nabla_{\theta} \mathcal{L}_{\psi}(\theta, \mathbf{d}^h), \mathbf{d}^r).$$

To learn from a video of a human, we need an adaptation objective $\mathcal{L}_{\psi}$ that can effectively capture relevant information in the video, such as the intention of the human and the task-relevant objects. While a standard imitation losses are applied to each time step independently, the learned adaptation objective must solve a harder task, since it must provide the policy with suitable gradient information *without* access to true actions. As discussed previously, this is still possible, since the policy is trained to output good actions during meta-training. The learned loss must simply supply the gradients needed to modify the perceptual components of the policy to attend to the right objects in the scene, so that the action output actually performs the right task. However, determining which behavior is being demonstrated and which objects are relevant will require examining multiple frames at the same time to determine the human's motion. To incorporate this temporal information, our learned adaptation objective therefore couples multiple time steps together, operating on policy activations from multiple time steps. Since temporal convolutions have been shown to be effective at

Figure 2: Example placing (left), pushing (middle), and pick-and-place (right) tasks, from the robot's perspective. The top shows human demos and the bottom shows robot demos.

processing temporal and sequential data (Van Den Oord et al., 2016), we choose to adopt a convolutional network to represent the adaptation objective $\mathcal{L}_\psi$, using multiple layers of 1D convolutions over time. Our algorithm is summarized in Algorithm 1 and Figure 1.

## 3 EXPERIMENTS

We run our experiments with a PR2 arm, with robot demonstrations collected via teleoperation and RGB images collected from a consumer-grade camera (unless noted otherwise). We compare the following meta-learning approaches: (1) a contextual policy that takes as input the robot's observation and the final image of the human demo and outputs the predicted action, (2) a DA-LSTM policy that directly ingests the human demonstration video and the current robot observation and outputs the predicted robot action, a domain-adaptive version of the algorithm by Duan et al. (2017), (3) our approach with a linear, per-timestep adaptation objective, and (4) our approach with the temporal adaptation objective. For measuring generalization, we use held-out objects in all of our evaluations that were not seeing during meta-training, and new human demonstrators. As illustrated in Figures 2 and 3, we consider three different task settings: placing a held object into a container while avoiding two distractor containers, pushing an object amid one distractor, and picking an object and placing it into a target container amid two distractor containers.

In our first experiment, we collect human demonstrations from the perspective of the robot's camera. For placing and pushing, we only use RGB images, whereas for pick-and-place, RGB-D is used. During evaluation, we collected one human demonstration per test object, and evaluated the policy inferred from the video. We report the results in Table 1. Our results show that, across the board, the robot is able to learn to interact with the novel objects using just

| One-shot success | placing | pushing | pick & place |
|---|---|---|---|
| DA-LSTM | 33.3% | 33.3% | 5.6% |
| contextual | 36.1% | 16.7% | 16.7% |
| DAML, linear loss | 76.7% | 27.8% | 11.1% |
| DAML, temporal loss (ours) | **93.8%** | **88.9%** | **80.0%** |

| One-shot pushing success | seen bg | novel bg 1 | novel bg 2 |
|---|---|---|---|
| DAML, temporal loss (ours) | **81.8%** | **66.7%** | **72.7%** |

Table 1: Top: placing, pushing, and pick-and-place, using human demos from the perspective of the robot. Bottom: pushing, using human demos with a different scene and camera. All use held-out objects and a novel human demonstrator.

one video of a human demo with that object, with pick-and-place being the most difficult task. We find that the DA-LSTM and contextual policies struggle, likely because they require more data to effectively infer the task. This finding is consistent with previous work (Finn et al., 2017b). Our results also indicate the importance of integrating temporal information when observing the human demonstration, as the linear loss performs poorly compared to using a temporal adaptation objective.

Now, we consider a challenging setting with human demonstrations collected in a different room with a different camera and camera perspective from that of the robot, as seen in Figure 3. We use a mounted cell-phone camera and ten different table textures. We consider the pushing task, as described previously. We evaluate performance on novel objects, a new human demonstrator, and with one seen and two novel backgrounds. The results for this experiment are summarized at the bottom Table 1. As seen in the supplementary video, we find that the robot is able to successfully learn from the

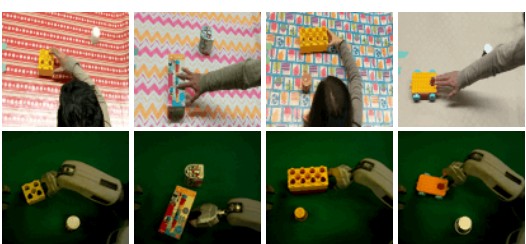

Figure 3: Human and robot demonstrations used for meta-training for the experiments with large domain shift. We used ten different diverse backgrounds for collecting human demonstrations.

demonstrations with a different viewpoint and background. Performance degrades when using a novel background, which causes a varied shift in domain, but the robot is still able to perform the task about 70% of the time. Videos of all results are at `sites.google.com/view/daml`

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
