# OpenReview forum: "One-Shot Imitation from Observing Humans via Domain-Adaptive Meta-Learning"
_ICLR.cc/2018/Workshop — Accept_

### Official Review · AnonReviewer3 · 2018-03-09
**Interesting look at handling domain shift via meta-learning and exploiting temporal information**

**Rating:** 7
**Confidence:** 3

**Review:**

The submission is looking at the problem of one-shot learning under domain shift, e.g. transferring from human to robot actions. The main novelties are the domain-adaptive extension of the MAML algorithm, and the introduction of a temporal adaptation objective.
For the inner adaptation objective, actions are not known, so the objective operates only on the policy activations. Therefore, meta-learning is required to provide a strong policy prior, such that the "fine-tuning" at inference time will output the right actions.

While the submission makes the above incremental contributions and regards a constrained set-up, experimental results are convincing, both numerically and in the provided video, and including ablation studies. In particular, using the proposed temporal loss makes a substantial difference under the presence of domain shift. The proposed approach is still fundamentally imitation learning of similar actions; I suspect that further approach adaptation are required to generalize further.
The 4-page workshop submission is unfortunately not as easy to read as the full paper, due to the necessity to condense.

---

### Official Review · AnonReviewer1 · 2018-03-13
**Great idea for learning how to imitate**

**Rating:** 10
**Confidence:** 4

**Review:**

The paper presents an algorithm for performing one-shot learning: visually imitate a human performing a task after having observed some pairs of demonstrations and imitations, potentially with different backgrounds, imitators or viewpoints.

Positive points:
The problem the paper is attacking is extremely challenging: how can we teach a robot to imitate in new conditions? The authors describe their approach in a clear manner with a good level of detail given the maximum length of the paper, and provide further details in the full version.
The authors offer some important insights, like defining the loss in terms of the policy activations instead of the actions (not available for the human demonstration). Or like using a an adaptation objective that exploits multiple frames from the demonstration.
The proposed approach is evaluated against two baselines, and ablated in terms of the contribution of the temporal loss. Moreover, to justify that there's indeed a non-trivial domain shift between train and test, the authors show that the performance doesn't degrade too much when observing imitations with a different viewpoint, background and demonstrator.

Negative points:
There are many things that could be added to the paper to make it more complete, like discussing more about how big the domain shift can be or how to attack the imitation of more complex actions. However, I think the available space for this workshop submission is well used and those details belong to a full submission of this work

---

### Decision · Program_Chairs · 2018-03-20
**ICLR 2018 Workshop Acceptance Decision**

**Decision:**

Accept

**Comment:**

Congratulations, your paper was accepted to the ICLR workshop.